# Pitfalls of Using Imaging Technique in the Presence of Eustachian Valve or Chiari Network: Effects on Right-to-Left Shunt and Related Influencing Factors

**DOI:** 10.3390/diagnostics12102283

**Published:** 2022-09-21

**Authors:** Han Zhang, Weiwei Liu, Jie Ma, Huanling Liu, Lin Li, Jinling Zhou, Shanshan Wang, Shanshan Li, Wei Wang, Yueheng Wang

**Affiliations:** 1Department of Cardiac Ultrasound, The Second Hospital of Hebei Medical University, Shijiazhuang 050000, China; 2Department of Vascular Surgery, The Second Hospital of Hebei Medical University, Shijiazhuang 050000, China

**Keywords:** patent foramen ovale, Eustachian valve, Chiari network, right-to-left shunt, intrapulmonary shunt, contrast transthoracic echocardiography

## Abstract

When patent foramen ovale (PFO) combines with the prominent Eustachian valve or Chiari network (EV/CN), contrast transthoracic echocardiography (cTTE) may miss the diagnosis of PFO. We sought to determine the characteristics of right-to-left shunt (RLS) in PFO patients with prominent EV/CN on cTTE and identify the causal factors of missed diagnosis. We consecutively enrolled 98 patients who suffered from PFO-related stroke and with prominent EV/CN. All patients were divided into the delayed and non-delayed groups according to the characteristics of RLS on cTTE. The characteristics of RLS were compared with those of 42 intrapulmonary shunt patients. The anatomical characteristics of PFO and EV/CN were analyzed in the 98 PFO patients. Upon cTTE, significantly delayed occurrence and longer duration of the RLS in the delayed group were found both at rest and during the Valsalva maneuver, similar to the intrapulmonary shunt. Multivariate logistic analysis revealed that the length of EV/CN (>19 mm) and the diameter of PFO at the left atrium aspect (<1.2 mm) were high-risk factors for missed diagnosis. In conclusion, RLS showed delayed emergence and disappearance in some of the PFO patients with prominent EV/CN. The length of EV/CN and the diameter of PFO may have been related to the missed diagnosis of PFO.

## 1. Introduction

The American Academy of Neurology recommends patent foramen ovale (PFO) closure in high-risk PFO patients with cryptogenic stroke to prevent stroke recurrence [1]. The prominent Eustachian valve or Chiari network (EV/CN) is one of the high-risk anatomic factors for PFO. It is not only associated with the cryptogenic stroke but also independently related to residual shunt and recurrent ischemic stroke even after PFO closure [2,3]. Therefore, accurate identification of PFO with prominent EV/CN is of great significance in decision-making in the management of the cryptogenic stroke.

Rigatell et al. showed that prominent EV/CN was associated with the spontaneous right-to-left shunt (RLS) and severe RLS in PFO patients [4,5]. However, previous studies have demonstrated that prominent EV/CN rendered PFO negative on contrast transthoracic echocardiography (cTTE) when the contrast agent was injected through an antecubital vein [6,7]. At present, there is no research to analyze the reasons for this contradiction. Hence, the present study sought to (1) observe the characteristics of RLS with prominent EV/CN on cTTE image and (2) identify independent risk factors that cause a missed diagnosis of PFO with prominent EV/CN.

## 2. Materials and Methods

### 2.1. Patient Population

Five hundred and six consecutive patients with PFO associated cryptogenic stroke were retrieved from August 2019 to August 2021 for this study. The patients were diagnosed with cryptogenic stroke by a multidisciplinary team according to the classification of the trial of Org 10172 in acute stroke treatment (TOAST) [8]. The exclusion criteria were two or more potential causes of stroke, incomplete evaluations, heart failure, atrial fibrillation, malignant tumor, severe hepatic and renal insufficiency, moderate or above valve regurgitation or stenosis, intrapulmonary shunt, congenital heart disease except PFO and inadequate provocative maneuver. All patients underwent cTTE and contrast transesophageal echocardiography (cTEE) examinations. The diagnostic criterion for PFO was RLS through PFO observed on cTEE [9]. A total of 124 patients were diagnosed with PFO combined with prominent EV/CN by cTEE. The patients with atrial septal defects (8 patients), intrapulmonary shunt (14 patients), and inadequate Valsalva maneuver (VM) (4 patients) were excluded, and 98 patients were included in the analysis (Figure 1). Forty-two patients with an intrapulmonary shunt were enrolled during the same study period; these patients all demonstrated without EV/CN. The diagnostic criterion for an intrapulmonary shunt was RLS through pulmonary veins observed on cTEE [10].

According to the three cardiac cycles rule, the PFO patients with prominent EV/CN were divided into the delayed and non-delayed groups. In the delayed group, RLS was delayed by >3 cardiac cycles after microbubbles filled the right atrium on cTTE, both at rest and after VM. Patients who could not meet the above conditions constituted the non-delayed group. The characteristics of RLS on cTTE in the delayed group, non-delayed group, and intrapulmonary shunt group were compared. The anatomical characteristics of PFO and EV/CN were analyzed in the delayed and non-delayed groups to identify independent risk factors for delayed characteristics of RLS on cTTE. 

### 2.2. Imaging and Measurements 

The Vivid E95 instrument (GE Vingmed Ultrasound AS, Horten, Norway) and second harmonic imaging were used for cTTE and cTEE. All patients were positioned in a left lateral position and underwent contrast echocardiography following the established guidelines of the American Society of Echocardiography [9]. For both cTTE and cTEE examinations, 8 mL saline, 1 mL air and 1 mL blood were pushed back and forth in two syringes for no less than 20 times to form agitated saline. Agitated saline was immediately injected into the cubital vein using a 20 G intravenous indwelling needle at rest and during the strain phase of VM. The VM was repeated for at least 3 times if microbubble did not appear or delayed in the left chambers.

The cTTE was performed by blowing a homemade pressure gauge during the provocative maneuver. The standard VM was performed to maintain a pressure of 40 mmHg for 10–15 s [11]. The cTTE image on the apical four-chamber view revealed the appearance and duration of microbubbles in left chambers after filling the right atrium both at rest and during VM. The quantification of RLS on cTTE was performed by counting the maximum number of microbubbles in the left chambers on one still frame, regardless of cardiac cycles [9].

A provocative maneuver during cTEE was indirect compression of the inferior vena cava [10,12]. We performed a manual compression of approximately 5 cm in depth, 5 cm to the right of the epigastric region for 30 s with subsequent release on entry of microbubbles into the right atrium. After relaxation, the atrial septum was immediately displaced to the left atrium. Prominent EV referred to an EV length of ≥10 mm and thickness of ≥1 mm [13]. EV length was measured from the muscular rim to the free-floating end in the right atrium in the bicasval plane. CN length was measured from the margin of the inferior vena cava or coronary sinus to the free-floating end in the right atrium in the bicasval plane. The diameters and tunnel lengths of PFO were measured during the release phase of the provocative maneuver. The PFO diameter was measured as the maximum separation between the septum primum and septum secundum, both at left atrium aspect and at right atrium aspect in the end-systolic frame (in the same view). The angles between the inferior vena cava and the PFO flap and between EV/CN and PFO flap were also measured. The presence of a left-to-right shunt was then assessed. At rest, the maximum mobility of the primary septum of >6.5 mm was defined as a hypermobile primary septum [14]. The cTEE study was performed on the second day after the initial cTTE study because of the requirement of fasting. 

### 2.3. Observer Variability 

The intraclass correlation coefficient was used to evaluate the intraobserver and interobserver variabilities using 15 randomly selected patients. Observer 1 and observer 2 repeated the measurements to assess the interobserver variability. Then, observer 1 determined new measurements to assess the intraobserver variability. All results were retrospectively analyzed by the two sonographers with no knowledge of the grouping.

### 2.4. Statistical Analysis 

All data analyses were performed using SPSS software (version 21.0, SPSS Inc., Chicago, IL, USA) and R version 4.0.3. Measurement data were expressed as mean ± standard deviation or the median (interquartile range). We used the Kruskal–Wallis H test to compare the characteristics of RLS on cTTE in the three groups. The count data were expressed as frequency and percentage, and the chi-square test and the Fisher exact test were performed. To permit nonlinear relationships, the cutoff potential for the length of EV/CN were explored in the restricted cubic spline model. Receiver operating characteristic curves were conducted to identify the cutoff value of the diameters of PFO for group classification. Univariate and multivariate logistic analyses were used to determine high-risk factors for the delayed characteristics of RLS on cTTE. All analyses were two-tailed, and a *p*-value of <0.05 was considered statistically significant. The cTTE results were analyzed for each patient, followed by the analysis of cTEE results.

## 3. Results

### 3.1. Characteristics of the Study Population

There were 20 (20.41%) and 78 patients in the delayed and non-delayed groups on cTTE, respectively. The baseline data of all patients are shown in Table 1.

### 3.2. Characteristics of RLS in PFO Patients with Prominent EV/CN on cTTE

The delayed group had significant delayed occurrence (*p* < 0.001) and longer duration (*p* < 0.001) of the RLS compared to the non-delayed group, both at rest and during VM (Table 2, Figure 2). The characteristics of RLS in the delayed group were similar to intrapulmonary shunt (Table 2).

### 3.3. Factors Related to RLS

The anatomical characteristics of PFO and EV/CN were analyzed in the delayed group and non-delayed group. There were 11 (55.00%) patients combined with EV and 9 (45.00%) patients combined with CN in the delayed group, against 51 (65.38%) patients combined with EV and 27 (34.62%) combined with CN in the non-delayed group. No significant differences in the distributions of EV and CN between the two groups were found. Restricted cubic spline showed a J-curved association between the length of EV/CN and the risk of delayed RLS (Figure 3). The risk of delayed RLS was relatively flat until around 19 mm of the length of EV/CN and then started to increase rapidly afterwards. According to the Receiver operating characteristic curves, the cutoff values of PFO diameter were 1.2 mm and 1.0 mm, respectively. Multivariate logistic analysis indicated that the length of EV/CN (>19 mm) and the diameter of PFO at left atrium aspect (<1.2 mm) were high-risk factors for the delayed RLS on cTTE (Table 3). We found that the superior vena cava blood containing agitated saline filled in the atrial septum during the diastolic phase. In the systolic phase, the inferior vena cava blood scoured microbubbles around the atrial septum (Figure 4, Appendix A).

### 3.4. Observer Variability

Table 4 shows the results of intraobserver and interobserver variabilities. All intraclass correlation coefficient values were >0.75, indicating the strong consistency.

## 4. Discussion

The main findings of this study can be summarized as follows: (1) RLS showed delayed emergence and disappearance in 20.41% of PFO patients with prominent EV/CN on the cTTE image, similar to intrapulmonary shunt. (2) The length of EV/CN (>19 mm) and the diameter of PFO (<1.2 mm) at the left atrium aspect were found to be the high-risk factors for the delayed features. (3) When the RLS was delayed and continued, a cTEE should be performed for differential diagnosis.

It has been confirmed that the prominent EV/CN was one of the high-risk factors for PFO associated cryptogenic stroke [1,2,3,4,5]. However, previous studies showed that when a contrast agent was injected via an antecubital vein, prominent EV/CN may interfere with the diagnosis of PFO on cTTE [6,7]. Schuchlenz et al. observed 31 PFO patients with prominent EV, which suggested that the sensitivity of cTTE was only 61% [7]. Our study found that the prominent EV/CN produced characteristics of RLS were similar to the intrapulmonary shunt and therefore rendered the missed diagnosis. The intrapulmonary shunt was characterized by a delayed appearance [15] and long duration [9]. The characteristics of a delayed shunt in PFO patients with persistent EV/CN could easily lead to the missed diagnosis of PFO in clinical practice. Hence, sonographers should wait for more cardiac cycles on cTTE after contrast injection. If there was a delayed RLS, not intrapulmonary shunt, the cTEE should be considered to confirm the interference of prominent EV/CN. Unlike a previous study [6], we found that cTEE was more helpful than cTTE in diagnosing PFO with prominent EV/CN. Chen et al. used cardiac catheterization or operation as a gold standard, which suggested that cTEE showed a significantly higher sensitivity than cTTE in diagnosing PFO (100% vs. 63%) [16]. We thought the cTEE was more sensitive because of its better anatomic exploration. An increasing number of guidelines used cTEE with appropriately performed provocative maneuvers as diagnostic criteria [10].

Importantly, our study found that the length of EV/CN (>19 mm) and the diameter of PFO at the left atrium aspect (<1.2 mm) were independent risk factors for the delayed features. We thought this could be explained by several factors. First, EV/CN could direct blood from the IVC into the left atrium through the PFO. A larger EV/CN could enhance the streaming effect and accelerate blood flow. Thus, the superior vena cava blood containing agitated saline filled defect at the atrial septum. Schuchlenz et al. confirmed that the color flow signal from IVC is better than the contrast medium from superior vena cava in the diagnosis of PFO on TEE, especially with the prominent EV/CN [7]. In addition, a larger EV/CN could produce the blocking effect. This may have been related to the extension of EV/CN in the diastole and contraction in the systole. Pagel et al. used TEE to demonstrate that CN stretched into the right chamber at the diastolic phase, and the blood of inferior vena cava moved slowly under the obstruction of CN, forming spontaneous contrast; the CN rolled into the inferior vena cava at the systolic phase, and blood flow of the inferior vena cava was accelerated, when the spontaneous contrast disappeared [17]. We could see the same phenomenon in the EV (Figure 4). Last, the smaller the PFO diameter, the lesser the probability of RLS through PFO. There was a significant positive correlation between the diameter of PFO and the amount of RLS [18]. Hence, a larger EV/CN and smaller diameter of the PFO could influence the time of appearance and disappearance of RLS.

Negative or delayed PFO-RLS may have been caused by an ineffective VM, left heart system disease, and respiratory movement [9,10]. In this study, both cTTE and cTEE images were examined by experienced sonographers, and the pressure gauges were used with the cTTE to quantify the VM. Atrial septum displacement to the left atrial side was observed during VM by the cTEE. The patients with increased left atrial pressure were excluded. In addition, patients in the delayed diagnosis group had fewer left-to-right shunts than those of the non-delayed group from the univariate analysis. Respiratory movements caused a delayed appearance of PFO-RLS due to discordance between the cardiac cycle and the respiratory rate, but the effect of respiratory movements was eliminated by VM [9]. However, this study found that PFO-RLS was still delayed in patients in the missed diagnosis group during VM.

The effect of long EV/CN on PFO-RLS may have led to the missed diagnosis of PFO on the cTTE image, which may have influenced the clinicians in choosing the right treatment for patients. Studies found that agitated saline through femoral vein injection increased the detection rate of PFO, especially in the presence of prominent EV/CN [19,20]. However, femoral venipuncture is not suitable for all patients screened for PFO, and complications such as arterial injury and hematoma formation may occur. This study suggested that the cTEE should be considered when there was a concern about the interference of prominent EV/CN. 

Some limitations of our work should be acknowledged. First, this study was a retrospective study. We adopted the continuous inclusion method to avoid selection bias. Second, the number of patients in this study was relatively small. The strict inclusion and exclusion criteria limited our ability to include a larger sample size. Therefore, a large-scale prospective study is required to confirm the results of this study.

## 5. Conclusions

The PFO with prominent EV/CN detected the delayed emergence and disappearance of RLS when the contrast agent was injected into the antecubital vein. The delayed phenomenon was related to the length of EV/CN and the diameter of PFO. The present study provided evidence that cTTE was confirmed as a valid tool for detecting PFO, since a RLS has been demonstrated in all cases. In doubtful cases with delayed and continued shunts, a cTEE should be performed for differential diagnosis.

## Figures and Tables

**Figure 1 diagnostics-12-02283-f001:**
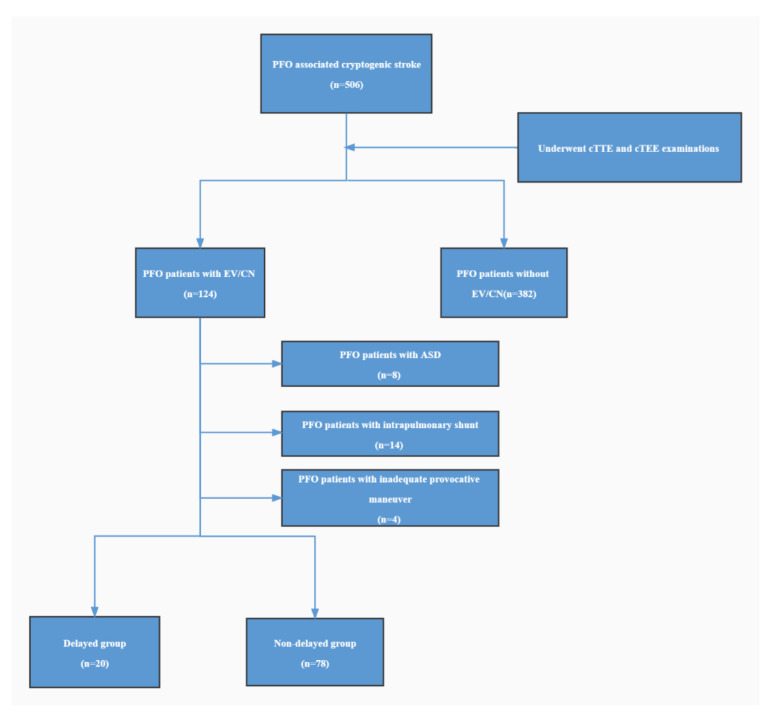
Flow chart. ASD = atrial septal defect; CN = Chiari Network; cTTE = contrast transthoracic echocardiography; cTEE = contrast transesophageal echocardiography; EV = Eustachian valve; PFO = patent foramen ovale.

**Figure 2 diagnostics-12-02283-f002:**
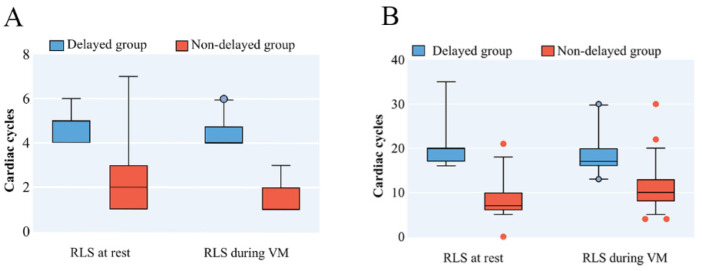
Characteristics of RLS in PFO patients with prominent EV/CN on cTTE. The delayed group had significant delayed occurrence (**A**) and longer duration (**B**) of the RLS compared to the non-delayed group, both at rest and during VM. CN = Chiari Network; cTTE = contrast transthoracic echocardiography; EV = Eustachian valve; PFO = patent foramen ovale; RLS = right-to-left shunt; VM = Valsalva maneuver.

**Figure 3 diagnostics-12-02283-f003:**
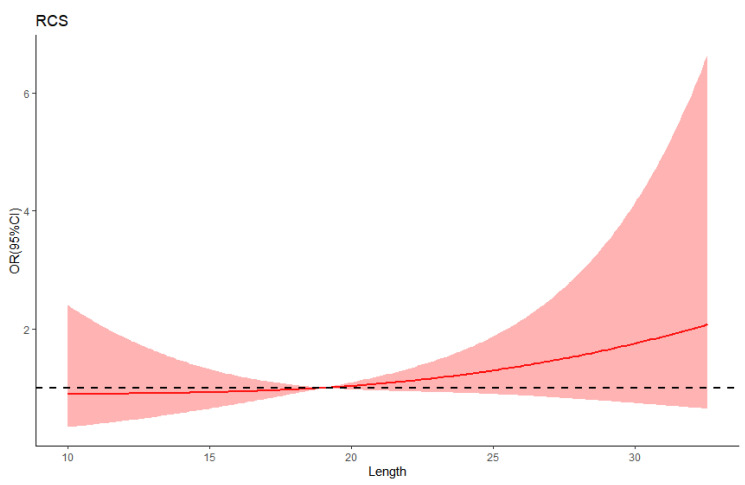
The relationship of the length of EV/CN and the risk of delayed RLS. The reference line for no association is indicated by a dashed black curve at a odds ratio of 1.0. The intersection of the dashed black curve and solid red curve represents the cut-off value of the length of EV/CN (19 mm). The light red area showing 95% confidence intervals derived from restricted cubic spline regressions. CN = Chiari Network; EV = Eustachian valve; RLS= right-to-left shunt; RCS= restricted cubic spline.

**Figure 4 diagnostics-12-02283-f004:**
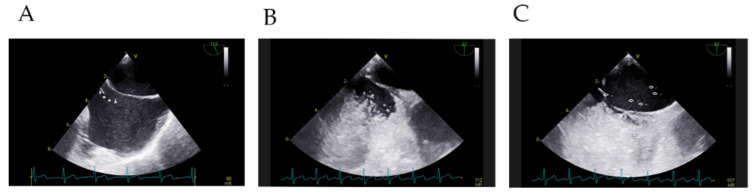
The cTEE images in a 34-year-old man with PFO. (**A**) shows the EV in the right atrium. The white asterisks indicate the outline of EV. (**B**) Due to the obstruction of the prominent EV during diastolic phase, microbubbles could not enter the left atrium through PFO. (**C**) The prominent EV rolled during the systolic phase; microbubbles could enter the left atrium through PFO. The white arrows indicate the microbubbles in the left atrium. cTEE = contrast transesophageal echocardiography; EV = Eustachian valve; PFO = patent foramen ovale (Appendix A).

**Table 1 diagnostics-12-02283-t001:** Patients Baseline Parameters.

Parameter	PFO with EV/CN Patients (*N* = 98)	PFO with EV/CN Patients		Pulmonary RLS (*N* = 42)	*p* Value
		Delayed Group (*N* = 20)	Non-Delayed Group (*N* = 78)		
Age, yrs	42 ± 14	35 ± 7 ***	46(31–55)	35 ± 11 ***	0.001 ^a^
Male	51(52)	13(65)	38(49)	14(33)	0.054
BMI, kg/m^2^	26(23–28)	25 ± 4	25 ± 3	24 ± 2	0.097
History of DM	8(8)	2(10)	6(6)	4(9)	0.915
History of HTN	24(24)	3(15)	21(27)	6(14)	0.206

Values are mean ± SD, *n* (%) or median (interquartile range). The *p* values refer to One Way ANOVA, Kruskal–Wallis H test, Chi-square. BMI = body mass index; DM = diabetes mellitus; HTN = hypertension; PFO = patent foramen ovale; EV/CN = Eustachian valve or Chiari network; RLS = right-to-left shunt. ^a^ Statistical significance (*p* < 0.05). ** p* < 0.05 vs. Non-delayed Group.

**Table 2 diagnostics-12-02283-t002:** Characteristics of RLS on cTTE.

Parameter	Non-Delayed Group	Delayed Group	Pulmonary RLS	*p* Value
Time of RLS Occurrence at Rest	2.00 (1.00–3.00)	5.00 (4.00–5.00)	5.00 (4.00–5.00)	<0.001 ^a^
Time of RLS Occurrence after VM	1.00 (1.00–2.00)	4.00 (4.00–4.75)	4.00 (4.00–5.00)	<0.001 ^a^
Duration at Rest	7.00 (6.00–10.00)	20.00 (17.00–20.00)	20.00 (20.00–25.00)	<0.001 ^a^
Duration after VM	10.80 ± 4.31	17.00 (16.00–20.00)	21.95 ± 5.99	<0.001 ^a^
Amount of RLS after VM	50.00 (31.00–79.75)	32.00 (26.25–45.50)	50 (31.75–100.00)	0.026 ^a^

Values are mean ± SD or median (interquartile range). The *p* values refer to Kruskal–Wallis H test. RLS = right-to-left shunt; cTTE = contrast Transthoracic Echocardiography; VM= Valsalva maneuver. ^a^ Statistical significance (*p* < 0.05).

**Table 3 diagnostics-12-02283-t003:** Univariate and Multivariate Logistic Analyses for Factors Related to Missing PFO in Patients with EV/CN (N = 98).

	Univariate Analysis		Multivariate Analysis	
Parameter	OR (95%CI)	*p* Value	OR (95%CI)	*p* Value
Prevalence of EV	1.545 (0.570–4.188)	0.392		
Length of EV/CN (>19 mm)	5.176 (1.585–16.906)	0.006 ^a^	8.412 (1.544–45.820)	0.014 ^a^
The diameter of PFO at left atrium aspect (<1.2 mm)	12.013 (3.221–44.804)	<0.001 ^a^	14.806 (2.415–90.780)	0.004 ^a^
The diameter of PFO at right atrium aspect (<1.0 mm)	12.000 (3.582–40.205)	<0.001 ^a^		
Length of tunnel	0.959 (0.833–1.103)	0.556		
The angle between IVC and PFO flap	1.014 (0.981–1.047)	0.421		
The angle between EV/CN and PFO flap	0.998 (0.973–1.024)	0.862		
Prevalence of LRS	0.436 (0.339–0.561)	<0.001 ^a^		
Hypermobile primary septum	0.859 (0.785–0.940)	0.166		
Age	0.955 (0.918–0.933)	0.02 ^a^		

CN = Chiari Network; EV = Eustachian valve; IVC = inferior vena cava; LRS = left-to-right shunt; OR = odds ratio; PFO = patent foramen ovale. ^a^ Statistical significance (*p* < 0.05).

**Table 4 diagnostics-12-02283-t004:** Intraobserver and Interobserver Variability of Characteristics of RLS.

	Outcomes Compared	ICC	95%CI
Appearance at Rest Intraobserver	O1(first outcome) vs. O1(second outcome)	0.951	0.864–0.983
Appearance at Rest Interobserver	O1(first outcome) vs. O2	0.951	0.864–0.983
Duration at Rest Intraobserver	O1(first outcome) vs. O1(second outcome)	1.000	1.000–1.000
Duration at Rest Interobserver	O1(first outcome) vs. O2	0.979	0.938–0.993
Appearance at VM Intraobserver	O1(first outcome) vs. O1(second outcome)	0.951	0.864–0.983
Appearance at VM Interobserver	O1(first outcome) vs O2	1.000	1.000–1.000
Duration at VM Intraobserver	O1(first outcome) vs. O1(second outcome)	0.971	0.917–0.990
Duration at VM Interobserver	O1(first outcome) vs. O2	0.945	0.845–0.981

The values are given as the ICC scores. ICC = intraclass correlation coefficient; O1 = observer1; O2 = observer2; VM = Valsalva maneuver.

## Data Availability

The data presented in this study are available on request from the corresponding author.

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
