# Peer review of "Pitfalls of Using Imaging Technique in the Presence of Eustachian Valve or Chiari Network: Effects on Right-to-Left Shunt and Related Influencing Factors"

_diagnostics, 2022, doi:10.3390/diagnostics12102283_

Round 1

Reviewer 1 Report

Missing a potentially important PFO due to an imaging bias is a well-known event and may preclude needed treatment to cryptogenic stroke patients. 

Indeed, contrast injection through the antecubital vein may yield false negative results due to the geometric relationship between the SVC and the atrial septum. This may be even more important in older patients with dilated and horizontal ascending aorta/Valsalva sinuses leading the SVC blood strema to mix with the IVC before entering the tricuspid valve or the PFO. The study is well planned and the statistical methods are exhaustive. 

I think the main messages are two: 1) wait for more cardiac cycles at cTTE after contrast injection and 2) if positive with a delay NOT due to intrapulmonary shunts tell the interventionalist he may have to deal with a long (>19mm) CN/EV. This aspect should be better emphasized by the authors. Moreover, do the authors think that all patients should be tested by cTEE instead of cTTE, because of the potential occurrence of false negative studies, ? Do they think the CN/EV issue is a limitation for the TCD technique as well?

The only real improvement I would suggest to the manuscript would be to put some more emphasis on the clinical impact of their data.

Overall, a comprehensive and accurate paper.

Reviewer 2 Report

The results and discussion were reasonable and well written.

If possible, Fig4 should be movie images.

Will the results of the manuscript change the current diagnostic cascade?

Both patients with positive contrastTTE, whether it occurred >3s or not, should  have TEE for the diagnosis.

The phenomenon in relation to Chiari network and Eustachian valve was interesting but it would not change the current diagnostic method.

Reviewer 3 Report

In the present paper dr Zhang and colleagues aim to detect anatomical factors contributing to potential pitfalls in diagnosis of PFO by contrast transthoracic echocardiography. 

For this purpose they analyse 98 patients with proven diagnosis of PFO at transesophageal echocardiography and identify 2 subgroups according to time of appearance of contrast into the left atrium

They demonstrate that a prominent Eustachian valve and a small PFO are independent predictors of late and reduced right-to-left shunt after saline contrast injection and, thus, can contribute to a lower diagnostic accuracy of TTE.  

The study is of significance to the field and adds new insights into PFO diagnostic work-up, as a late appearance of microbubbles in the left chambers might represent a misleading factor, favouring a  diagnosis of intrapulmonary shunt. 

The paper is well written and easy to read. 

My comments are listed below: 

Methods: Please define indirect IVC compression

I would replace EV and CN “diameter” with “length”

Since at multivariate analysis PFO diameter is a crucial  determinant of missing diagnosis its assessment should be described more in detail, even with the help of a figure 

Results: Were RLS characteristics similar between TEE and TTE in terms of delay and  duration?

Discussion: The sentence “Last, smaller diameter damaged the RLS though the PFO” is not clear. Please rephrase

Table 1: which group value refers to? Could the authors please be clearer

In the figure 4B microbubbles into the left atrium are not clearly visualized. Could the authors please indicate LA bubbles with arrows or asterisks?

Conclusions: I believe that the key point of the present study is that TTE is confirmed as a valid tool for detecting PFO, since a right-to-left shunt has been demonstrated  in all cases.

In doubt cases, with delayed and transitory shunt, a TEE should be performed for differential diagnosis

These considerations could be highlighted in the conclusions

The reference 19 is not correctly stated
